# Blind Trading: A Literature Review of Research Addressing the Welfare of Ball Pythons in the Exotic Pet Trade

**DOI:** 10.3390/ani10020193

**Published:** 2020-01-22

**Authors:** Jennah Green, Emma Coulthard, David Megson, John Norrey, Laura Norrey, Jennifer K. Rowntree, Jodie Bates, Becky Dharmpaul, Mark Auliya, Neil D’Cruze

**Affiliations:** 1World Animal Protection 222 Gray’s Inn Rd., London WC1X 8HB, UK; BeckyDharmpaul@worldanimalprotection.org (B.D.); NeilDCruze@worldanimalprotection.org (N.D.); 2Ecology & Environment Research Centre, Department of Natural Sciences, Manchester Metropolitan University, Manchester M1 5GB, UK; E.Coulthard@mmu.ac.uk (E.C.); D.Megson@mmu.ac.uk (D.M.); J.Norrey@mmu.ac.uk (J.N.); lauranorrey@googlemail.com (L.N.); J.Rowntree@mmu.ac.uk (J.K.R.); Jodie.Bates@stu.mmu.ac.uk (J.B.); 3Zoological Research Museum Alexander Koenig, Department Herpetology, Adenauerallee 160, 53113 Bonn, Germany; mark.auliya@ufz.de; 4Department of Conservation Biology, Helmholtz Centre for Environmental Research GmbH—UFZ, 04318 Leipzig, Germany; 5Wildlife Conservation Research Unit, Department of Zoology, University of Oxford, Recanati-Kaplan Centre, Tubney House, Abingdon Road, Tubney, Abingdon OX13 5QL, UK

**Keywords:** exotic pet, *Python regius*, welfare domains, health, wildlife trade

## Abstract

**Simple Summary:**

The Ball python is a small species that is commonly kept as an exotic pet across the world. Despite huge numbers of these snakes being kept and traded in the pet industry, there is very little information available about how catching, breeding, transporting and housing them in captivity could impact their welfare. Our study reviewed the published literature for this species and found 88 relevant peer-reviewed scientific papers. Physical health was the predominant focus of research, with numerous studies reporting on disease, injury or clinical treatments. Far fewer papers focused on other aspects of Ball python wellbeing, including behaviour, nutrition, environment or mental condition. We also found that very few studies focused on wellbeing prior to pet ownership, i.e., during the early stages of the trade chain when they are caught from the wild, transported, or bred in captivity. We recommend that more research is needed to assess the impact of the exotic pet trade on this species’ welfare. In particular, research on welfare conditions during capture and transportation of wild Ball pythons, and the potential effects of captive breeding, could help reduce suffering throughout the trade.

**Abstract:**

Extensive numbers of Ball pythons are caught, bred, traded and subsequently kept in captivity across the world as part of the exotic pet industry. Despite their widespread availability as pets, relatively little is known about the potential welfare challenges affecting them. We reviewed the literature for research focused on the health and welfare of Ball pythons in the international pet trade. From a total of 88 articles returned from the search criteria, our analysis showed that very few actually focused on trade (10%) or animal welfare (17%). Instead, the majority (64%) of articles focused on veterinary science. There was a considerable bias towards physical health, with most studies neglecting the four other domains of animal welfare (behaviour, nutrition, environment and mental health). Furthermore, very few studies considered Ball pythons prior to resulting pet ownership, during wild capture and transportation or captive breeding operations. Our review demonstrates that our current understanding of welfare for Ball pythons traded as exotic pets is limited. We recommend that future research should focus on aspects of the industry that are currently overlooked, including the potential consequences of genetic selection during captive-breeding and the conditions provided for snakes prior to and during international transportation.

## 1. Introduction

Ownership of non-domesticated animals, or ‘exotic pets’, has become increasingly popular across the world [1]. The exotic pet industry is a substantial part of the global trade in wildlife products, which is worth an estimated $30.6–42.8 billion USD annually [2]. Reptiles comprise a substantial part of the live animal trade (>20%) [3] and are particularly prevalent as exotic pets in European and North American markets [3,4,5,6], for example, where around 0.9 million are kept in UK homes and 9.4 million in US homes [1,7,8]. The scale of the trade is likely to be even greater than current estimates due to incomplete record-keeping and widespread illegal activity throughout the industry [1,9]. Increasing consumer demand for novel colour and pattern strains produced by artificial breeding selection is driving industry growth for the captive breeding sector [10].

Ball pythons (*Python regius*) are one of the most common species of reptiles kept as exotic pets, dominating in trade volume [3,6]. From 1978 to 2017, between 0.9 million and 1.6 million live individuals were exported from Togo alone, 99% of which were for commercial purposes (presumably as exotic pets) [11]. The emergence of novel colour and pattern strains known as ‘morphs’ has also been a significant driver of growth in demand for captive bred Ball pythons by creating competition among breeders and owners for the most unusual and exotic characteristics [10]. Their popularity has been attributed in part to their docile nature, long lifespans and small size, deemed by some as suitable for terrariums [12,13]. Consequently, Ball pythons are often referred to as great ‘beginner’ exotic pet snakes that are relatively easy to care for in captivity [14,15].

Animal welfare refers to the wellbeing of non-human animals and is described by the American Veterinary Medical Association as the state of an animal in relation to the conditions in which it lives. There are a range of animal welfare challenges associated with the private ownership of reptiles, including Ball pythons, as exotic pets [16,17,18]. For example, Ball pythons have specific requirements regarding diet, lighting, hygiene, space, temperature and humidity [19]. Yet in a recent review involving more than 5000 individual Ball pythons in North America and Europe, D’Cruze et al. [20] found that the entities involved in this commercial enterprise were not providing housing conditions that meet the minimum welfare recommendations, either in public or privately, for periods of time that could range from several days to many years. The same study found that vendors selling Ball pythons online and at pet expositions were not providing husbandry guidance for new owners. The consequences of failing to meet these requirements or provide adequate information on how to do so can negatively impact reptile welfare, resulting in disease, injuries, stress-related behaviours [21], mental suffering and mortalities associated with poor husbandry. In addition, intense breeding selection of gene mutations to create novel morphs leads to inbreeding, resulting in genetic disorders with consequential health issues [20].

Ball pythons also face a multitude of animal welfare challenges during the trade chain prior to international export for private ownership as exotic pets. For wild caught and ranched animals, methods of capture and transportation can incite high levels of stress and physical injury [21]. Ball pythons are ranched when their eggs are taken from the wild and reared on farms. Once hatched, a proportion of individuals are returned to the wild, and the rest remain on the farm to be used commercially. Wildlife farms have been criticised for crowded or unhygienic conditions [22] that can potentially cause disease, suffering, as well as aggression or harassment from other co-occupants and competition for vital resources, such as water [23,24]. Exact mortality rates prior to and during transportation are unknown and could be significant [6,25]. Unintended harm resulting from poorly managed wild release of ranched animals (e.g., genetic pollution and disease) is also of potential concern [26].

Despite their widespread availability as exotic pets, we still know relatively little about captive reptile welfare. There is a widely acknowledged bias towards mammalian species in the scientific literature [27,28,29]. This bias extends to welfare research, where publications concerning the welfare of non-mammalian species are vastly outnumbered [30]. Herpetofauna, despite making up 46% of species richness of terrestrial vertebrates [31], are neglected in research on cognition, sentience, enrichment of captive environments and other areas of work concerning welfare prioritised in mammals and birds [28,32,33]. The lack of attention reptiles receive in comparison to other taxa has limited our understanding of their sensory abilities and functional requirements. The more we learn about reptiles, the more apparent the potential deficiencies associated with their lives in captivity become [34].

The aim of this study is to review the existing scientific literature for research focused on the health and welfare of Ball pythons in the international pet trade. We are focusing on the scientific literature, rather than associated grey literature, to examine the evidence base that knowledge throughout the industry relies on. The extensive numbers of this species that are being caught, bred, traded and kept in captivity across the world arguably warrants a thorough understanding of the potential adverse consequences associated for Ball pythons as exotic pets. In addition to searching the literature for research investigating trade, we review research pertaining to their health using search terms relating to disease and welfare. An improved understanding of the potential welfare challenges Ball pythons could experience in the exotic pet industry is imperative given the mounting physiological, neuroanatomical and behavioural evidence that reptiles are sentient beings, capable of suffering [33].

## 2. Materials and Methods

### 2.1. Literature Search

We conducted a systematic review of the scientific literature. A total of 26 search terms relating to health, welfare and trade were used (disease, pathogen, virus, viral, bacteria, bacterial, parasite, parasitic, fungus, fungal, health, welfare, exotic pet, trade, capture, transport, captive care, captive breeding, behaviour, husbandry, suffering, nutrition, diet, pain, harm, distress). Each search term was employed with the Boolean operator ‘AND’, with three additional terms (Ball python, Royal python, *Python regius*). Searches were conducted for the time period 2009–2019. Across all term combinations, 78 different searches were employed in total. This was repeated across three journal databases (PubMed(Bethesda, USA); Scopus (Amsterdam, The Netherlands); Web of Science (Philadelphia, USA). Google scholar was excluded because it returned large numbers of non-relevant literature.

### 2.2. Literature Analysis

Of the 130 papers returned from the literature search, 16 could not be sourced due to institutional access issues. A further 26 did not actually relate to Ball pythons, (only mentioned them in reference to other research) and were therefore removed from the dataset. Two papers relating to ‘Python’ software rather than animals were also removed from the returned list. This left a total of 88 papers remaining, which were included in the analysis. The literature was analysed by recording five different aspects of the content: focus, target words, use, welfare domains and pathogens.

Focus: Papers were categorised in terms of their primary focus (animal welfare, conservation, veterinary science and wildlife trade, see Appendix B), and country of origin of the study listed (where no location was given for study, the country of first author was used).

Target words: Each paper was searched for five target words, related broadly to reptile welfare: ‘welfare’, ‘suffering’, ‘pain’, ‘distress’ and ‘harm’.

Use: In reference to those papers focused specifically on wildlife trade, use of snakes in relation to the paper were categorised as ‘kill on site’, ‘capture, transport live and kill for use’ or ‘live use’.

Welfare domains: The dataset was also analysed with regards to mention of the five domains of animal welfare, a systematic assessment framework devised to assess animal’s welfare state through consideration of positive and negative experiences [35]. The experiences are split into five categories: four physical domains (environment, nutrition, physical health and behaviour) and one mental domain. Papers were recorded if they mentioned food and water deprivation, environmental challenge or discomfort, disease or injury, behavioural restriction or anxiety and stress.

Pathogens: Any mention of ‘bacteria’, ‘fungi’, ‘parasite’, ‘protozoa’ or ‘virus’ were noted as a result of searching the document. All disorders, diseases or conditions were recorded in relation to Ball pythons, with a list of specific named pathogens and parasites compiled. In addition, any recommendations made by the authors were collated.

### 2.3. Statistical Analysis

All analysis was carried out in R version 3.6.1 (R Core Development Team, 2019). Chi-square goodness of fit tests were used to investigate the distribution of papers published across research category and across locations. Results were recorded as the percentage of papers.

## 3. Results

Our results are derived from a literature analysis of 88 relevant peer-reviewed scientific papers, returned from our search criteria.

Focus: Figure 1A shows the percentage of papers on Ball pythons in each of the assigned primary research focus categories. There was a significantly uneven split across these categories (χ^2^ = 71.36, df = 3, *p* < 0.001), with ‘veterinary science’ (64%) being the most common focus followed by ‘animal welfare’ (17%) (Figure 1A). The location of study was also not evenly distributed (χ^2^ = 198.84, df = 20, *p* < 0.001), with the USA having the largest percentage of papers (32%), followed by Denmark, Germany, Italy, France and Poland (Figure 1B). None of the studies originated in West Africa, where Ball pythons are the most common legally exported species.

Target words: Our target word searches of the 88 papers found that 4% mentioned the word “welfare” (*n* = 4), only 1% mentioned “suffering” (*n* = 1) and 12% mentioned one of the following stress related terms: “pain”, “distress” or “harm” (*n* = 14).

Use: Details of Ball Python use in the studies concerned showed that no papers relating to wildlife trade indicated snakes were killed on sight, or indicated snakes were captured, transported and then killed for use and 77% referred specifically to live use (*n* = 55). The other 23% of papers did not specify use.

Welfare domains: Figure 2 shows the percentage of papers that identified each of the five welfare domains. Of the 88 papers, 51% (*n* = 45) considered negative aspects of the ‘health’ domain, citing disease or injury. Within this domain, the majority of issues raised were physical illnesses (diseases, parasites, etc.), rather than behavioural issues or injury (burns, bites, etc.) (Appendix B; Appendix D
Appendix A). Far fewer papers mentioned the four remaining domains. Only 8% (*n* = 7) of the studies addressed the ‘environment’ domain, citing challenges and discomfort arising from the animal’s surroundings, such as the effects of inappropriate temperature and humidity. Another 7% (*n* = 6) addressed negative aspects of the ‘behaviour’ domain, concerned with abnormal behaviours such as open mouth breathing, head tremors and lethargy. Negative experiences in the ‘mental state’ domain, such as anxiety, fear and distress were mentioned in 6% (*n* = 5) of the studies. Finally, ‘nutrition’ was the least cited domain, where only 3% (*n* = 3) of the papers referred to deprivation of food and/or water, and incorrect nutrition from an inadequate diet.

Pathogens: The majority of the pathogens reported were bacteria and parasites, with fewer instances of protozoa, fungi and viruses (Figure 3; Appendix D). Details of the specific health and behavioural issues related to Ball python welfare indicated by the papers are provided in Appendix C and Appendix D (with associated definitions provided in Appendix A).

## 4. Discussion

Our study provides the most comprehensive review of published literature addressing the welfare of Ball pythons in the exotic pet trade chain, to date. A total of 88 articles were returned from our search criteria, although very few specifically addressed trade (10%) or animal welfare (17%). Rather, the vast majority of articles were focused on veterinary science, describing over 100 clinical symptoms and nearly 150 underlying pathogens (Appendix C; Appendix A), many of which were artificially imposed on the snakes during controlled experiments. In the context of the five domains of animal welfare, our analysis showed a considerable preference towards physical health. Around two thirds of the studies mentioned disease and injury. In contrast, fewer than 10% of published literature referred to the snake’s environment, nutrition, behaviour or mental wellbeing. Furthermore, none of the studies were conducted in, or referred to welfare and health aspects in West Africa, where Ball pythons are the most numerically exported species for use as pets listed on CITES.

### 4.1. Animal Welfare Domains

The relatively extensive coverage of the physical condition and associated clinical signs observed in captive Ball pythons could benefit their welfare as it can help enable appropriate veterinary attention and changes to husbandry practices when necessary. However, focusing the majority of research in one area while neglecting other key domains of animal welfare could have negative implications for Ball pythons. For example, only six studies (7%) looked at the snake’s behaviour, even though behavioural change in reptiles can often serve as the primary indicator of disturbance, injury or disease [24]. Behavioural assessments may also offer the advantage of detecting subclinical or psychological conditions, such as under stimulation from their environment, that may not be revealed through physiological measurement [24]. Informed behavioural assessment of reptiles could therefore be a valuable and non-invasive means of evaluating welfare [36,37]. Thus far, this appears to have been underutilised in the assessment of Ball pythons in the exotic pet trade chain.

Similarly, many negative health and welfare experiences are a consequence of poor nutrition or an inadequate captive environment [34]. A poor diet can cause a range of diseases, and malnutrition and dehydration have been linked to abnormal shedding or dysecdysis in snakes [38]. Inappropriate lighting, hygiene, space, temperature and humidity can all cause or exacerbate health conditions [34]. Despite the importance of these factors, only nine papers (9%) considered these two domains collectively. The fifth domain, which addresses mental experiences, was only mentioned in five articles (6%) in the published literature. As wild animals, Ball pythons may not have evolved coping mechanisms to adequately deal with all of the artificial stressors presented in a given captive environment [24,39,40], which in some cases could be at odds with the innate adaptations of the species. Consideration of mental stresses such as states of fear and frustration are necessary to consider a broad view of the animal’s experience and to ensure a holistic assessment of their welfare [9].

### 4.2. The Trade Chain

In particular, studies concerning welfare during the early stages of the trade chain are currently under-represented in the scientific literature (i.e., during wild capture, ranching and transportation). For wild caught and ranched snakes, crowded transport, unhygienic conditions, poor nutrition, poor environmental control, poor handling and aggression from co-occupants are just some of the factors that could cause suffering [25]. However, no welfare assessments of snake ranching can be found in the existing published scientific literature, and welfare during transportation appears to be a completely unrecorded issue [6]. Currently it is unclear whether there are any welfare provisions for the snakes prior to CITES and International Air Transport Association (IATA) regulations which come in to force during international border crossings. This is particularly concerning because the limited data available suggest mortality rates for reptiles in transit could be as high as 33% [6,41].

It is not only the welfare of wild caught and ranched snakes that requires further scientific attention. To date, very little formal research has been undertaken to understand the health and welfare consequences of selectively breeding wild sourced and captive reptiles [10]. Anecdotal reports of potential welfare issues linked to genetic manipulation can be found in hobbyist media (e.g., duckbills [42]), but only one study in the published scientific literature focuses on a single genetic disorder (wobble head syndrome) associated with a widely propagated phenotype of the Ball python [10]. For most recently established and ‘trendy’ morphs, there is not yet enough data to draw conclusions. However, as the ‘novelty’ of new morphs appears to have been prioritised over research into the health consequences of this inbreeding, it is possible that some new diseases and disorders resulting from selective breeding could be reported in the future.

Our review suggests there is insufficient literature to fully assess the welfare impacts of the trade in Ball pythons as exotic pets. This raises questions about the ethics of the industry as a whole; is it humane for us to continue trading and keeping animals without a comprehensive understanding of their experience? A group of key UK veterinary organisations proposed greater controls on the exotic pet business by only approving species for the pet trade where the animal’s needs have been fully researched and understood, and where there is reasonable evidence from published literature and professional experience that these needs can be met [43]. Warwick et al. [1] describe ‘positive lists’ a similar principle of using evidence-based methods to determine the suitability of species for trading and keeping. The proposal of positive lists has generally been well accepted among veterinarians and allied professionals and is in development processes across Europe and North America [1]. The information currently available for Ball pythons would likely be insufficient to justify their place in the exotic pet market, using the positive list principle.

### 4.3. Limitations

It is the nature of reviews such as this one that, as a result of the target words and journal databases used, some relevant articles may be missed. However, given our extensive yet balanced application of 26 different target words in relation to health, welfare and trade across three different journal databases, this review provides a comprehensive and useful representation of the existing scientific literature focused on the health and welfare of Ball pythons in the international pet trade. Furthermore, we recognise that information about the captive care of Ball pythons is also widely available in the hobbyist media and grey literature, and that our study could be considered limited by only including scientific literature in our analysis. However, the aim of our study was to assess the evidence base that advice and knowledge throughout the industry is based on. We believe that highlighting the scarcity of formal publications demonstrates that whatever other information exists for pet owners is likely based on a lack of scientific evidence. This raises questions about what premise local regulations and advisory standards of care are based on, and whether the ownership of Ball pythons as exotic pets can be considered as ‘responsible’. Furthermore, grey literature and hobbyist media can be subject to non-scientific ‘folklore husbandry’ [44,45], a term coined to describe methods of husbandry deemed ‘best practice’ without proper evaluation, justified by the notion that ‘it has always been done that way’ [44]. Therefore, by eliminating these types of sources we can be sure that the information gathered in our review is scientifically valid.

### 4.4. Recommendations

It is hoped that the evidence presented in this review can be used to improve welfare for captive Ball pythons. Acknowledging research gaps in the published scientific literature and focusing research towards previously neglected areas of the industry could help improve captive conditions and practices for snakes kept and traded for the exotic pet industry. More informative research would also be valuable to aid the implementation of policies that mitigate or minimise harm during the trade chain. In the case of the Ball python, this would particularly apply to regulations governing captive breeding practices in the USA and Europe, and wild capture and ranching methods in key source countries (i.e., Togo, Benin and Ghana). 

Although broader research could improve captive conditions for Ball pythons, it is important to consider that the only way to fully mitigate all of the potential negative animal welfare impacts for Ball pythons is to remove them from the pet trade all together. This would require changes in legislation in conjunction with education and awareness campaigns to inspire behavioural changes among exotic pet owners and breeders. Schuppli et al. [25] argued that the wide-spread popularity of keeping non-traditional pet species renders it impractical to try to end exotic pet ownership and trade. Nevertheless, popularity should not be considered justification for an industry to continue if it is associated with animal suffering. In the absence of removing Ball pythons from the pet trade all together, any improvements to conditions and practices would be advantageous to the millions of snakes living within the industry.

## 5. Conclusions

Our review highlights that research addressing Ball pythons in the exotic pet trade chain is limited within the published literature. Moreover, the sparse information that does exist is focused on a few specific facets, neglecting several animal welfare domains and key stages of the trade chain that supports this commercial enterprise. This paucity of information could hinder efforts to safeguard the welfare of the animals involved.

From the limited literature available it is already clear there are a multitude of welfare issues associated with different aspects of the trade chain. The lack of research addressing welfare throughout each stage, particularly during captive breeding, ranching and transport, is likely limiting our awareness of the negative consequences that the exotic pet industry can have on Ball pythons. More importantly, millions of animals are currently experiencing these welfare threats first-hand. Consequently, we recommend particular focus should be placed on the potential consequences of selective captive-breeding for ‘rare’ morphs and conditions provided for snakes prior to and during international transportation, where there is currently a dearth of knowledge. It is hoped this study demonstrates that our current understanding of welfare for Ball pythons traded as exotic pets is superficial and would benefit from a broader range of research throughout the industry.

## Figures and Tables

**Figure 1 animals-10-00193-f001:**
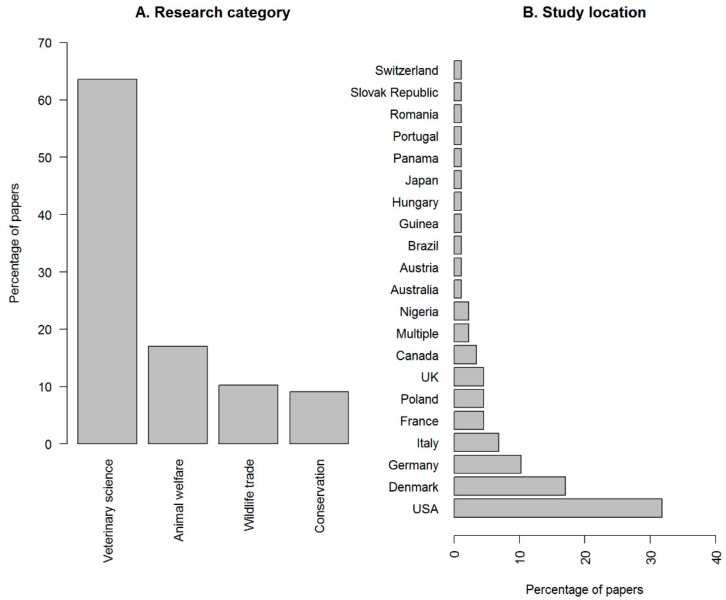
Percentage of papers per research focus category (**A**), and by location of study (**B**). Total number of papers included *n* = 88.

**Figure 2 animals-10-00193-f002:**
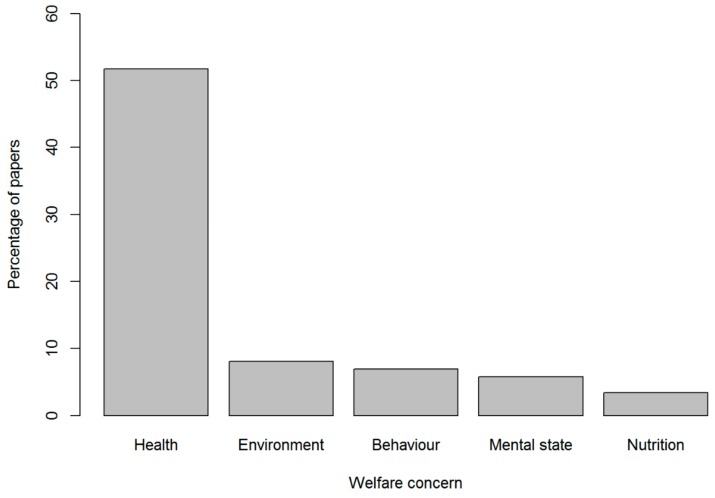
Percentage of papers that identified each of the five welfare domains.

**Figure 3 animals-10-00193-f003:**
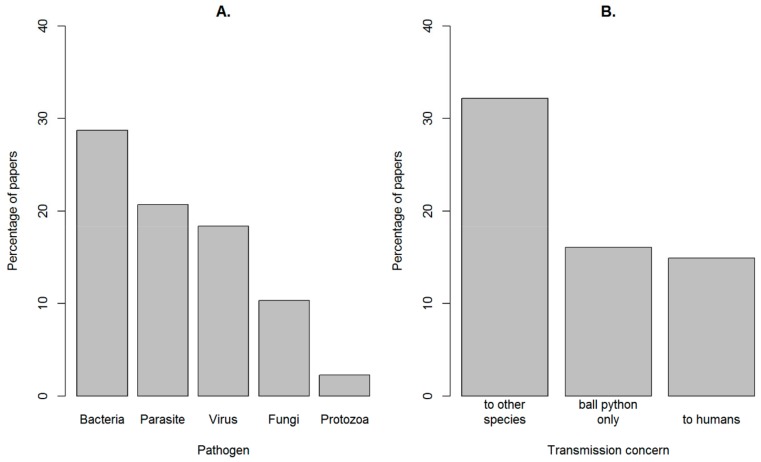
Percentage of papers that mentioned different pathogen types (**A**) and the different transmission concerns (**B**). A full list of the specific pathogen species (or taxa) is shown in Appendix D.

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
