# Peer review of "Blind Trading: A Literature Review of Research Addressing the Welfare of Ball Pythons in the Exotic Pet Trade"

_animals, 2020, doi:10.3390/ani10020193_

Round 1

Reviewer 1 Report

This manuscript represents an interesting review of peer-reviewed literature on the welfare of ball pythons. I found the manuscript to be well-written and well-referenced, and found relatively few grammatical mistakes/typos (see below for a comprehensive list). The methodology is clear and concise. Overall, I did not find any major issues or problematic sections that require extensive rewriting or revision.

Generally speaking, there are some words and phrasing used that could be better worded to give less a less 'biased' outlook/perspective, such as changing "suffering" to "reduced welfare" or something along those lines. While I personally feel that grey literature can offer some great insight into captive management that can be applied within the framework of welfare and should not be overlooked when making management decisions about the husbandry of any species, the authors do sufficiently explain their reasoning for omitting these publications from their study. I do wonder how their results would have differed had they included relevant grey literature.

General comments

Below are some more specific comments, suggestions and questions that could help improve the manuscript if addressed.

Lines 56-57: Although admittedly unfamiliar with these studies and their methodologies, I would say that these numbers appear to be off since the trade in reptiles in the USA is far greater than it is in the UK. The numbers quoted may be due to the limitations of those studies - it caught my eye and is something that may intrigue other readers familiar with these trades.

Line 67 - Although many people may argue that small enclosures (e.g., "rack systems") are acceptable housing for ball pythons because these animals can live and breed in them, an argument can also be made that smaller enclosures are not suitable for these animals because they fail to accommodate a number of important biological factors such as proper thermal gradients that allow full-body thermoregulation, or the ability of an animal to stretch out completely. Thus, smaller enclosures may have important welfare implications. Clifford Warwick and colleagues' recent 2019 paper on spatial considerations of snakes that you cite discusses this topic at length.

Line 84 - It would be useful to briefly explain the difference between wild-caught and ranched animals, since it seems as though these two concepts are often confused/misinterpreted in the hobby/trade.

Line 217 - The word "symptom" should be changed to "sign", as symptoms are not experienced by non-human animals. Symptoms refer to a condition or physical state that is apparent to the patient. Since this does not apply to animals, such conditions/states that are observed by clinicians and keepers are "signs", not "symptoms".

Line 227 - a useful study to cite here would be:

Eagan, T., 2019. Evaluation of Enrichment for Reptiles in Zoos. Journal of applied animal welfare science, 22(1): 69-77.

Line 228 - one could argue that behavioral indicators of welfare have been underutilized for all reptiles, not just ball pythons.

Line 228 - something to consider (but not necessarily add to the ms) - could this lack of behavioral assessment be related to the way in which this species is often maintained - in dark, opaque rack systems that prevent the keeper from viewing the animal?

Lines 238-240 - One could argue that the cognitive abilities of reptiles in general have been poorly studied, and underestimated historically

Line 246 - Consider changing "suffering" to "reduced welfare" or a less biased phrase/term.

Line 250 - Perhaps you could mention studies on reptile mortality in reptile dealers' facilities to further emphasize the need for studies on welfare during transport. For example, Ashley et al. (2014) found that some 40+% of reptiles died within 10 days of arriving at a dealer's facility. Full citation:

Ashlety S, Brown S, Ledford J, Martin J, Nash AE, Terry A, Tristan T, Warwick C. 2014. Morbidity and mortality of invertebrates, amphibians, reptiles and mammals at a major exotic companion animal wholesaler. Journal of Applied Animal Welfare Science 17: 308-321.

Lines 263-265. In regards to the authors' questioning of whether it is humane to continue keeping certain species without a comprehensive understanding of their care/welfare, I would raise the question - are there any reptile species that truly meet this criteria? If the authors are suggesting that more information is needed, how is the reptile keeping community supposed to develop a comprehensive understanding of care/welfare without first keeping and learning from these animals? What other options are there? Conversely, if the authors are advocating a complete ban on the keeping of reptiles, then they should be clear, open and up-front about this.

Lines 265-268 - If such standards were set/enacted, this would essentially eliminate the trade in all species, since there are currently insufficient welfare data available for all reptile species kept in captivity. I do not see how such information on welfare can be obtained without experimenting with conditions/practices and learning from species maintained in captivity.

Line 273 - Perhaps a brief explanation of the 'positive list' principle can help readers have a better idea of what it is beyond being similar to ensuring the needs of a species have been researched and understood.

Lines 278-280 - It might be useful to introduce/refer to the concept of "folklore husbandry" here. See:

Arbuckle K (2013) Folklore husbandry and a philosophical model for the design of captive management regimes. Herpetol Rev 44:448–452

Mendyk RW (2018) Challenging folklore reptile husbandry in zoological parks. In Berger M, Corbett S (eds), Zoo animals: husbandry, welfare and public interactions, Nova Science Publishers, Hauppauge, p 265–292

Grammatical errors/typos

Line 24 - delete "on" from "could impact on their welfare"

Line 33 - suggest changing language from "suffering" to "reduced welfare", etc.

Lines 36-37 - not sure what "with captive this industry" is supposed to mean.

Line 52 - change "is" to "has become"

Line 53 - insert "the" in between "part of" and "global trade"

Line 79 - delete "on" in between "impact" and "reptile welfare"

Line 89 - insert "and" in between "unknown" and "could"

Line 119 - add "the" in between "with" and "Boolean"

Line 152 - "test" should be plural - "tests"

Line 231 - Dysecdysis should be lower-cased: dysecdyssis

Line 282 - missing period (.) after "responsible".

Author Response

Thank you for your comments and insights. We have incorporated your feedback and we hope these changes have strengthened our manuscript. Please see the attached word document for detailed responses to your comments and suggestions. 

Reviewer 2 Report

Brief Summary: This paper offers the first literature review of scientific studies related to Ball python welfare in the exotic pet trade. While captive animal welfare research remains predominantly mammal-focused, scholars have begun to turn their attentions to the physical and mental well-being of reptiles housed in captivity. This paper is important because it highlights that our scientific understanding of the welfare threats to Ball pythons across four of the five welfare domains and in all stages of the exotic pet industry is currently limited. As such, it calls for additional research into Ball pythons' welfare and brings the ethicality of the entire pet trade into question. However, I have one major concern and several additional comments that I believe may limit the potential impact of this important work. 

Major Concern:

The authors used many more biological health search terms  (disease, pathogen, virus, viral, bacteria, bacterial, parasite, parasitic, fungus, fungal, health) than welfare terms (just the word welfare) and no conservation or nutrition terms, and then found more biologically relevant papers than welfare, conservation, or nutrition papers.  Should we be surprised by this?  Isn’t it just a reflection of the search performed and not some gap in the literature?

Broad comments 

Strengths: The paper is written in a clear, logical, and straightforward way, and it is potentially of interest to animal welfare advocates, policy-makers, and reptile industry professionals in addition to scholars of animal welfare. The methods are easily repeatable, and it makes an important contribution to the literature on captive reptile welfare by highlighting the lack of attention paid to a heavily traded, and greatly understood, species. By calling into question the ethicality of the entire Ball python pet trade, the authors make it clear that their main concern is the well-being of the animals in question, and not the economic benefits of the pet trade. This paper has the potential to inspire much-needed welfare studies of Ball pythons, and to encourage future literature reviews of welfare studies targeted at other reptile species.

Weaknesses: There are several minor inconsistencies in the writing, outlined in the specific comments section below. Furthermore, there are several sections where the authors might consider elaborating on some of the statements they have made and terms they have used in the discussion and methods section, as some terms may be unfamiliar to non-experts and other sentences appear to be too vague without elaboration. Again, specific instances of vagueness are included in the specific comments section below. Finally, the conclusion section emphasizes the importance of increasing our knowledge of python welfare for the human benefit of improving our understanding of the species. However, it is critical to note the other, perhaps more important benefit of increasing our knowledge of python welfare for the sake of the snakes themselves, as the animals that are experiencing the welfare threats first-hand and suffering as a result of the pet trade industry. 

Specific comments 

Line 40: 56% is not the “vast” majority.   

Lines 56 and 62: In line 56 .9 million and .4 million are used, but in line 62 963,344 and 1,657,814 are used; it would be helpful to keep all numbers written in a consistent manner.

Lines 78-80: The sentence "The consequences of failing to meet these requirements or provide adequate information on how to do so can negatively impact on reptile welfare, resulting in disease, injuries and stress-related behaviours." You might consider including mental suffering as well as stress-related behaviors, to emphasize that reptiles are capable of experiencing these mental states.

Line 121: Why these odd timeframes (2014-2019; 2009-2019)?  Please clarify.

Line 133: Please specify that the "country of origin" refers to the country of origin of the study, and not the Ball pythons; this was not initially clear.

Line 135: Key words already means something else, so a different way to describe your methods would be helpful here.  Also, you state there are four welfare keywords, but you list five of them. Before listing the five keywords, it would also be helpful to specify that they are related to welfare. In other words: "Each paper was searched for 5 key words, related broadly to reptile welfare."

Lines 158-163 (focus section of results): Although you elaborate on it in the discussion section, it is worth noting in the results that none of the studies originated in West Africa, where pythons are most commonly exported.

Lines 173-175 (use section of results): Can you explain what the remaining 23% of wildlife trade papers mentioned about python use? Did they mention nothing, or did they mention another type of use that was not originally searched for?

Lines 176-185 (welfare domains section of results): Please elaborate on some of the terms and findings of this section. Specifically, what is meant by "behavioural restriction," and does this include abnormal behaviours? Please also specify what is meant by challenges and discomfort arising from the animal's surroundings; it's not clear whether this means physical or social surroundings, and what types of surroundings would cause discomfort (a tank that is too small? Not enough greenery provided?). Finally, did the papers consider incorrect nutrition/diet or overfeeding in addition to food and water deprivation?

Line 238: I recommend removing the term "primitive" when describing mental capacities of reptiles. Even if fear and frustration may be considered primitive states, there is a danger that using such terms, even in a paper emphasizing reptile welfare, may be interpreted to mean that pythons are only primitive animals and unworthy of such welfare considerations. No descriptive term is needed in this sentence.

Line 270: Please briefly define 'positive lists' so it is clear to an unfamiliar reader.

Lines 298-300: Consider removing the line about Schuppli's claim regarding the infeasibility of trying to end the trade of exotic animals due to its "popularity." Just because an industry is popular does not justify letting it continue. It would suffice to simply lead into the next sentence without including Schuppli's comments.

Author Response

(The authors gave the same response as above.)

Reviewer 3 Report

In this manuscript, “Blind trading: a literature review of research addressing the welfare of Ball pythons in the exotic pet trade,” the authors categorize literature on Ball pythons based on the percentage of papers relating to veterinary science, animal welfare, wildlife trade, and conservation. They also report the percentage of papers originating from various countries. Importantly, they find that of the 5 domains of animal welfare, the vast majority of papers discuss physical health, and far fewer relate to environment, behavior, mental state, or nutrition. The manuscript also reports the types of pathogens and the transmission concerns (i.e., to pythons, to humans, or to other species) found in the literature.

The paper does not define welfare anywhere- the authors could develop a paragraph or two defining this term and describing how to measure it, and why it is important.

There are a few comments for the authors:

Line 21: there is a disagreement between singular/plural; the sentence should be “The Ball python is a small species that is commonly kept as an exotic pet…”

Line 36: grammatical change suggested: “…about the potential welfare challenges affecting them.”

Line 45: “future research should focus ON aspects…”

Line 56: “where, FOR EXAMPLE, around 0.9 million…”

Line 90: does wild release of ranched pythons happen regularly?

Line 104: please rephrase the sentence “the evidence base that knowledge is based on”

Line 121: why were these two time periods used, and why do they overlap?

Line 147: Here you say all disorders/diseases/conditions were all recorded in relation to pythons, but the definitions provided in the supplementary tables are taken from a human medical dictionary and need to be revised in some cases. Additionally, one of your figures indicates the transmission concern from python to humans, and python to other species, so have you differentiated those pathogens from the ones listed here?

Line 194: see comments on definitions in supplementary materials, below

Figure 3, panel B: above you say that diseases are defined relative to pythons, though this is not the case as some are defined for humans. Perhaps reorganize the appendix in terms of transmission concern (as in the figure) so that each disease can be defined for the correct species grouping?

Line 278: devil’s advocate: isn’t a lot of knowledge gained through experience (ownership), not necessarily through scientific testing? Is there anything we, as scientists, could learn from hobbyists?

Line 283: perhaps you should suggest future studies/research aimed at debunking folklore; as I suggested above, perhaps information from grey literature and hobbyist media is not incorrect because it was not learned through rigorous scientific testing per se.

Comments on supplementary material:

Table 1: behavior

Please define terms as related to pythons: what are their abnormal postures? Can they become disoriented without drugs? Can pythons be described as “lazy” or “indifferent” in a definition of lethargy? Or “absentminded” to describe stargazing?

Table 2: health

As all the definitions came from a human medical dictionary, many terms need to be rewritten to apply to pythons, and several others which cannot be found in humans (eg dysecdysis) need to be defined in the first place. In terms of formatting, it is not necessary to repeat the term being defined in the second column; for items with multiple terms, please try to define cohesively—that is, not defining each term separately.

Some definitions, as they have been taken from human medical dictionary, may not apply to pythons, eg “segmented epidermal erosion” has a definition referring to the erosion of the uterine cervix and loss of hard substance of a tooth.

Tables 3-6: is it really necessary to define bacteria, parasite, protozoa, and viruses? Wouldn’t a list suffice? I don’t think it adds anything to the paper to know the shape of a particular bacteria, for example.

Author Response

(The authors gave the same response as above.)

Reviewer 4 Report

It is an interesting article highlighting the gap in the scientific literature about reptile welfare and more specifically ball pythons in the exotic pet trade. Nevertheless, it seems that there is a systematic bias in this study due to the terms selected in the literature search. 11 out of the 14 terms selected in the literature search focus on pathogens/health while only 3 ("welfare", "exotic pet", "trade") focus on non-health issues. This selection of terms could explain why veterinary articles were predominantly found. I suggest to include keywords related to animal husbandry and capture (such as capture, captive care, captive breeding, behaviour, husbandry, nutrition) in the analysis before concluding and highlighting that several animal welfare domains are neglected in the literature.

Author Response

(The authors gave the same response as above.)

Round 2

Reviewer 2 Report

The authors did a very good job with the revisions and I am satisfied with their responses. 

I would only note that there is still an imbalance in the search terms and this will cause some readers to qualify the results.  One option that I think the authors should seriously consider is to code ALL the papers on ball pythons.  According to my search, there are only 211 papers in Web of Science that contain "ball python" or "Python regius", only 80ish more papers than what they have already coded.  I think looking at these additional papers would be unlikely to change the results or conclusions, but would allow the authors to make the strongest point possible regarding the gaps in the literature and thus substantially elevate the impact of the paper.    

Reviewer 4 Report

The authors addressed my comments.
